# Human papillomavirus type 38 alters wild-type p53 activity to promote cell proliferation via the downregulation of integrin alpha 1 expression

**Maria Carmen Romero-Medina**[1☯], **Assunta Venuti**[1☯], **Giusi Melita**[1¤a], **Alexis Robitaille**[1], **Maria Grazia Ceraolo**[1¤b], **Laura Pacini**[1¤c], **Cecilia Sirand**[1], **Daniele Viarisio**[2¤d], **Valerio Taverniti**[1], **Purnima Gupta**[1¤e], **Mariafrancesca Scalise**[3], **Cesare Indiveri**[3], **Rosita Accardi**[1], **Massimo Tommasino**[1] *

**1** International Agency for Research on Cancer (IARC), World Health Organization, Cours Albert Thomas, France, **2** Deutsches Krebsforschungszentrum (DKFZ), Im Neuenheimer Feld, Heidelberg, Germany, **3** Unit of Biochemistry and Molecular Biotechnology, Department DiBEST (Biologia, Ecologia, Scienze della Terra), University of Calabria, Arcavacata di Rende, Italy

☯ These authors contributed equally to this work.
¤a Current address: Giusi Melita, Laboratory of Cell and Gene Therapy "Stefano Verri", Tettamanti Research Center, ASST-Monza, San Gerardo Hospital, Monza, Italy;
¤b Current address: Maria Grazia Ceraolo, Tumor Immunology Unit, Division of Immunology, Transplantation and Infectious Diseases, San Raffaele Scientific Institute, Milano, Italy;
¤c Current address: Laura Pacini, Division of Molecular Pathology, The Institute of Cancer Research, London, United Kingdom;
¤d Current address: Daniele Viarisio, Freelance;
¤e Current address: Purnima Gupta, Duve Institute Université Catholique de Louvain–UCLouvain, Louvain-la-Neuve, Belgium.
* tommasinom@iarc.fr

**Data Availability Statement:** All relevant data are within the manuscript and its Supporting

## Abstract

Tumor suppressors can exert pro-proliferation functions in specific contexts. In the beta human papillomavirus type 38 (HPV38) experimental model, the viral proteins E6 and E7 promote accumulation of a wild-type (WT) p53 form in human keratinocytes (HKs), promoting cellular proliferation. Inactivation of p53 by different means strongly decreases the proliferation of HPV38 E6/E7 HKs. This p53 form is phosphorylated at S392 by the double-stranded RNA-dependent protein kinase PKR, which is highly activated by HPV38. PKR-mediated S392 p53 phosphorylation promotes the formation of a p53/DNMT1 complex, which inhibits expression of integrin alpha 1 (*ITGA1*), a repressor of epidermal growth factor receptor (EGFR) signaling. Ectopic expression of *ITGA1* in HPV38 E6/E7 HKs promotes EGFR degradation, inhibition of cellular proliferation, and cellular death. *Itga1* expression was also inhibited in the skin of HPV38 transgenic mice that have an elevated susceptibility to UV-induced skin carcinogenesis. In summary, these findings reveal the existence of a specific WT p53 form that displays pro-proliferation properties.

Information files. Raw data of all IB experiments are uploaded on Figshare (DOI: 10.6084/m9. figshare.12733427).

**Funding:** The study was supported by a grant from Fondation ARC pour la recherche sur le cancer (no. PJA 20151203192) (https://www.fondation-arc. org/espace-chercheur) and the Institut National de la Santé et de la Recherche Médicale (no. ENV201610) (https://www.eva2.inserm.fr/EVA/jsp/ AppelsOffres/CANCER/) to MT. The funders had no role in study design, data collection and analysis, decision to publish, or preparation of the manuscript.

**Competing interests:** The authors have declared that no competing interests exist.

## Author summary

This study shows that beta HPV38 can convert p53 functions from a tumor suppressor to an oncoprotein via the formation of a transcriptionally repressive complex, which in turn represses *ITGA1* expression, promoting cellular proliferation and UV-induced skin carcinogenesis.

## Introduction

Cellular transformation is intimately linked to alteration of pathways regulated by tumor suppressors, leading to reprogramming of cellular gene expression. Key negative regulators of cellular transformation are the products of two tumor suppressor genes, *TP53* and retinoblastoma (*pRb*).

p53 has the classic features of a transcription factor, containing a transactivation domain and a DNA binding domain (DBD) [1, 2]. The DBD is able to recognize specific response elements (REs) located within promoters of a vast number of cellular genes, including those activated by cellular insults, leading to cell-cycle arrest and/or apoptosis [1, 2]. The functions of p53 are tightly regulated by post-translational modifications, including methylation, acetylation, and phosphorylation [1, 2].

pRb negatively regulates cellular gene expression by interacting with several transcription factors, for example members of the E2F family, including E2F1–3 proliferation [3]. Upon exposure of quiescent cells to mitogenic signals, pRb is phosphorylated by cyclin-dependent kinase complexes, losing its ability to interact with E2Fs, which, in turn, are free to activate the expression of many genes that encode proteins with pro-proliferative functions [3]. In agreement with their negative role in cellular proliferation, loss of functional p53- and pRb-regulated pathways occurs in all cancer cells. In approximately 50% of human cancers, p53 is inactivated by mutations in the DBD or alternatively by post-translational mechanisms or synthesis of p53 variants with dominant-negative functions of p53 [1, 2]. Similarly, the functions of pRb are altered by deletions or mutations within its gene or, more frequently, by activation of cellular mechanisms that promote its hyper-phosphorylation, resulting in activation of E2Fs and unscheduled proliferation [3]. Interestingly, studies in *in vivo* experimental models highlighted the concept that E2F1 may also display tumor suppressor functions, inducing apoptosis [4–6]. E2F1 post-translational modifications, such as methylation, appear to influence its property of promoting cellular proliferation or apoptosis [7, 8].

In addition to inactivation of p53 and pRb, cellular transformation is associated with other events, including alteration of the integrin network [9]. Integrins are cellular receptor proteins that bind to and respond to the extracellular matrix [9]. In vertebrates, 24 integrin heterodimers have been identified, which display different substrate specificity and tissue expression [10]. Importantly, different studies provide evidence that certain integrins can act as promoters of tumorigenesis, whereas others may act as tumor suppressors [9]. In addition, the same integrin, as observed for E2F1, appears to exert an oncogenic or tumor suppressor function depending on the genetic background of the different cancer cells [9].

Thus, the available data not only underline the importance of inactivation of pathways controlled by tumor suppressors in human cancers, but also highlight the concept that some cellular proteins may have different functions in different cancer cells, promoting or inhibiting proliferation or transformation. For p53, it is well demonstrated that certain mutations in the DBD could confer oncogenic gain-of-function activities of the encoded product [11, 12]. In

contrast, very little is known about the biological properties of the wild-type (WT) p53 form, which is present in approximately 50% of human cancers.

Alterations of p53-, pRb-, and integrin-regulated pathways are also key events in cellular transformation induced by oncogenic viruses, such as human papillomaviruses (HPVs) (reviewed in [13] [9, 14–16]). Mucosal HPV types belonging to genus alpha are referred to as high-risk HPV types and are the etiological agents of several types of human cancers. In addition, beta HPV types, together with UV radiation, appear to be involved in the development of cutaneous squamous cell carcinoma (cSCC) [17–20].

Using *in vitro* and *in vivo* experimental models that express E6 and E7 oncogenes from cutaneous beta HPV type 38 (HPV38), we have characterized novel properties of a WT p53 form that exerts pro-proliferation functions via inhibition of integrin alpha 1 (*ITGA1*) expression.

## Results

### HPV38 deregulates integrin gene expression

We have previously shown that beta HPV38 E6 and E7 immortalize primary HKs [21, 22], altering p53- and pRb-regulated pathways. To evaluate whether HPV38 E6 and E7 are able to alter the integrin network, we determined the expression levels of different integrin genes in immortalized HPV38 E6/E7 HKs (38HK) that were transduced with recombinant retrovirus expressing the two viral genes (Fig 1A). As a control, we used primary HKs transduced with the empty retroviral vector pLXSN (HK). We observed that the expression levels of *ITGA1* and *ITGB8* were significantly downregulated by the viral oncoproteins (Fig 1A). Interestingly, it was previously shown that ITGA1 is implicated in negative regulation of epidermal growth factor receptor (EGFR) signaling and cellular proliferation [23]. Moreover, *ITGA1* downregulation has been associated with poor patient outcome and drug resistance in ovarian cancer [24]. Therefore, we focused our study on *ITGA1* and further validated its downregulation in 38HK by using TaqMan PCR. Also with this assay, we observed a statistically significant decrease in *ITGA1* mRNA levels compared with HK (Fig 1B). In addition, immunoblotting (IB) showed that 38HK has lower ITGA1 protein levels than HK (Fig 1C). In contrast, accumulation of p53 is induced by the viral proteins as previously shown (Fig 1C) [25].To evaluate whether the decrease in *ITGA1* mRNA levels is a direct consequence of the viral gene expression and is not due to the immortalization of 38HK, we used as an experimental model primary HKs expressing the human telomerase reverse transcriptase (*hTERT*) gene, which extends the life span of primary cells. *ITGA1* mRNA levels were significantly reduced in hTERT HKs expressing HPV38 E6 and/or E7 genes compared with mock-treated cells (Fig 1D). In addition, the inhibition of *ITGA1* gene expression appears to be associated mainly with E6 protein (Fig 1D). IB confirmed that the expression of HPV38 E6 and E7 decreases the ITGA1 protein levels also in hTERT HKs (Fig 1E). Similarly, immunofluorescence experiments showed the downregulation of *ITGA1* in hTERT HKs expressing HPV38 E6 and E7 genes (Fig 1F). Together, these findings show that HPV38 targets the integrin network.

### p53 and DNMT1 form a complex that is recruited to the *ITGA1* promoter in 38HK

To gain insights into the mechanism by which HPV38 deregulates *ITGA1* expression, we performed an analysis of the *ITGA1* promoter to identify putative REs using TFBind and JASPAR software. The analyses revealed the presence of several p53 REs in a region upstream of the transcriptional start site, spanning from the -936 to -835 nucleotides (S1A Fig). We have

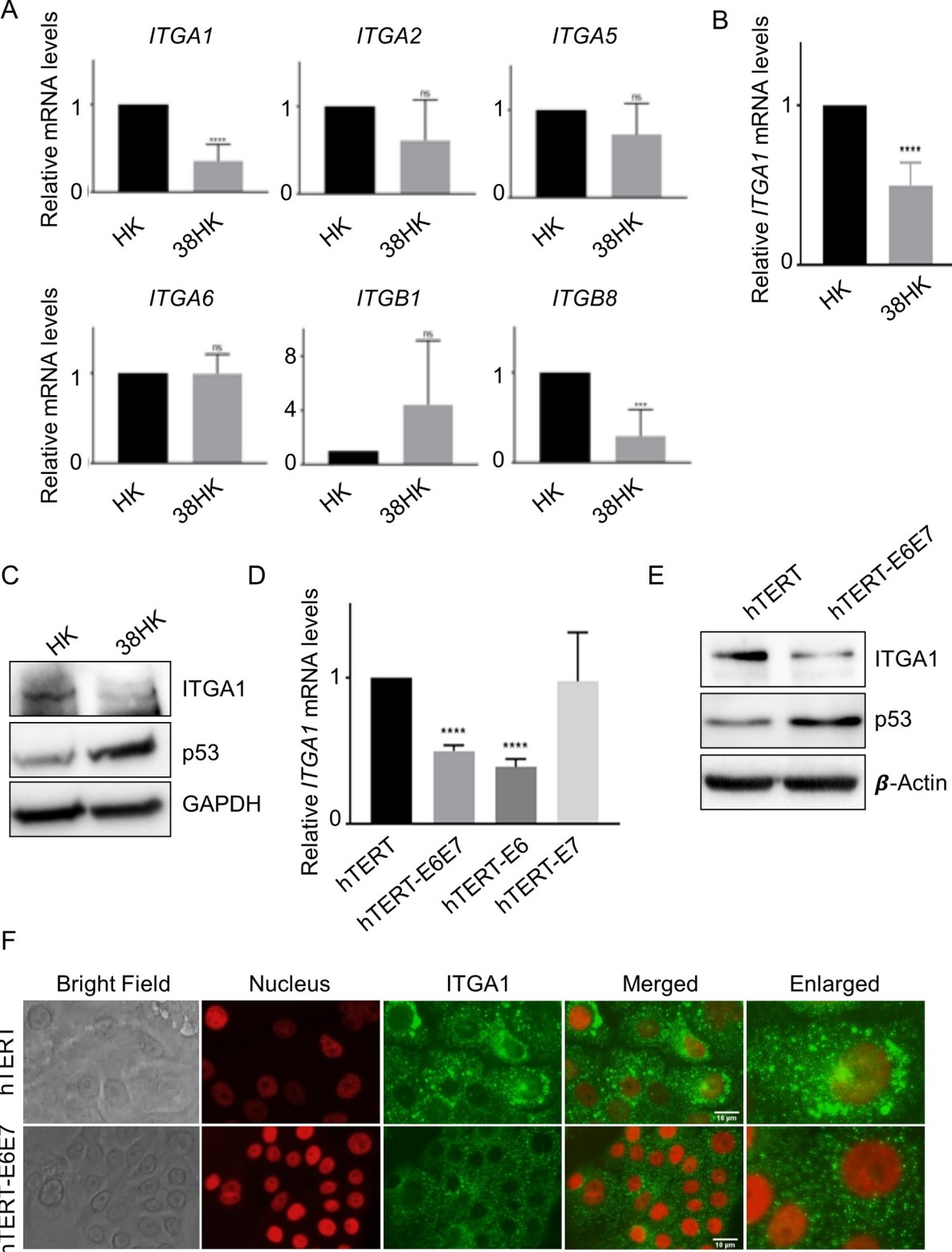

**Fig 1. *ITGA1* expression is downregulated in HPV38 E6/E7-expressing cells.** (A) Primary HKs were transduced with pLXSN HPV38 E6/E7 or pLSXN. mRNA levels were measured by RT-qPCR and normalized to *GAPDH*. Error bars represent standard deviations from 3 biological replicates of 2 different donors ($n = 6$). ***, $p<0.001$; ****, $p<0.0001$; ns, not significant. (B) Total RNA levels of HKs expressing or not expressing HPV38 E6 and E7 were analyzed by TaqMan PCR. Commercial probes for *ITGA1* and *GAPDH* were used. Results were

normalized to *GAPDH*. Data shown are the means of 3 independent experiments for 2 different donors ($n = 6$). ****, $p < 0.0001$. (C) Proteins extracts from HKs expressing or not expressing HPV38 E6 and E7 were analyzed by immunoblotting (IB) with the indicated antibodies. (D) The TaqMan assay was also performed as previously described in primary HKs previously retrovirally transduced with the *hTERT* gene and expressing E6 and/or E7 from HPV38 ($n = 3$). Results were normalized to *GAPDH*. ****, $p < 0.0001$. (E) Proteins extracts from hTERT pLXSN or hTERT HPV38 E6/E7 cells were analyzed by IB with the indicated antibodies. Images shown are representative examples of 2 different experiments. (F) hTERT pLXSN or hTERT HPV38 E6/E7 cells were plated on coverslips and after 24 h were probed for ITGA1 using anti-ITGA1 antibody followed by secondary Alexa Fluor 488-conjugated antibody. Nuclei were stained with DAPI (pseudocoloured red), and cells were analyzed under a microscope. Images were merged using ImageJ software.

previously shown that 38HK have high levels of a specific form of p53, which is phosphorylated only at two serines (15 and 392) [25]. To evaluate whether these putative p53 REs have the ability to interact with p53, we performed an electromobility shift assay using oligos encompassing the WT or mutated REs. RE2 showed a strong signal for p53 binding that was highly reduced upon mutation of the p53-binding motif (Fig 2A, lines 2 and 5) or by competition with WT unlabeled probe, but less efficiently with mutated unlabeled probe (Fig 2A, lines 7–9). These data were corroborated by chromatin immunoprecipitation (ChIP) experiments that showed a significant enrichment on RE2 compared with the negative control (Fig 2B). Recruitment of p53 to *ITGA1* promoter was not observed in HK (Fig 2C). Oligo pulldown experiments using biotinylated DNA probes, which contain a region of the *ITGA1* promoter encompassing RE2, revealed that p53 was efficiently precipitated by RE2 together with the epigenetic enzyme DNMT1, which is known to be associated with gene expression silencing (Fig 2D). DNMT1 recruitment to the *ITGA1* promoter was also confirmed by ChIP experiments (Fig 2E).

Inhibiting p53 functions by using the chemical inhibitor pifithrin severely impaired the interaction of p53 and DNMT1 with the *ITGA1* promoter (Fig 2F), indicating that these two cellular proteins are part of the same complex. ChIP-reChIP experiments confirmed their interaction and their recruitment to p53RE2 of the *ITGA1* promoter (Fig 2G). Immunoprecipitation (IP) experiments with p53 antibodies confirmed the interaction with the two cellular proteins (Fig 2H). Similarly to what has been observed with p53 inhibition by pifithrin, silencing the expression of *DNMT1* by siRNA significantly affected the recruitment of p53 (Fig 2I). Together, these data show that p53 and DNMT1 form a complex, and their interaction appears to be important for the binding to p53RE2 of the *ITGA1* promoter.

## The p53/DNMT1 complex inhibits *ITGA1* expression

Next, we evaluated the impact of the p53/DNMT1 complex on the transcription of the *ITGA1* gene. Treatment of 38HK with the p53 inhibitor pifithrin resulted in a 2-fold increase in *ITGA1* mRNA levels (Fig 3A). Similarly, in 38HK transient transfection experiments, we observed that the knockdown of *p53* using CRISPR/Cas9 technology correlated with an increase in *ITGA1* mRNA levels (Fig 3B). In agreement with mRNA levels, IB showed that p53 CRISPR/Cas9 also increased ITGA1 protein levels (Fig 3C).

Several isoforms of p53 have been identified with truncations at the N- or C-terminus and altered transcriptional functions [26]. To determine whether the inhibition of *ITGA1* expression is mediated by the full-length p53 form, we overexpressed it with the HA-tag at the N- or C-terminus in 38HK. In both cases, we observed by IB that the full-length p53 form is recognized by the HA-tag antibody (Fig 3D). Both full-length p53 fusion proteins further repressed the expression of *ITGA1* in 38HK (Fig 3E).

We have shown above that silencing of DNMT1 expression resulted in a loss of recruitment of the p53/DNMT1 complex to the *ITGA1* promoter (Fig 2G). Accordingly, *ITGA1* mRNA levels increased upon inhibition of *DNMT1* expression (Fig 3F). Similarly, treatment with 5-aza-2′-deoxycytidine, which is a 2′-deoxycytidine analogue and is a global demethylating

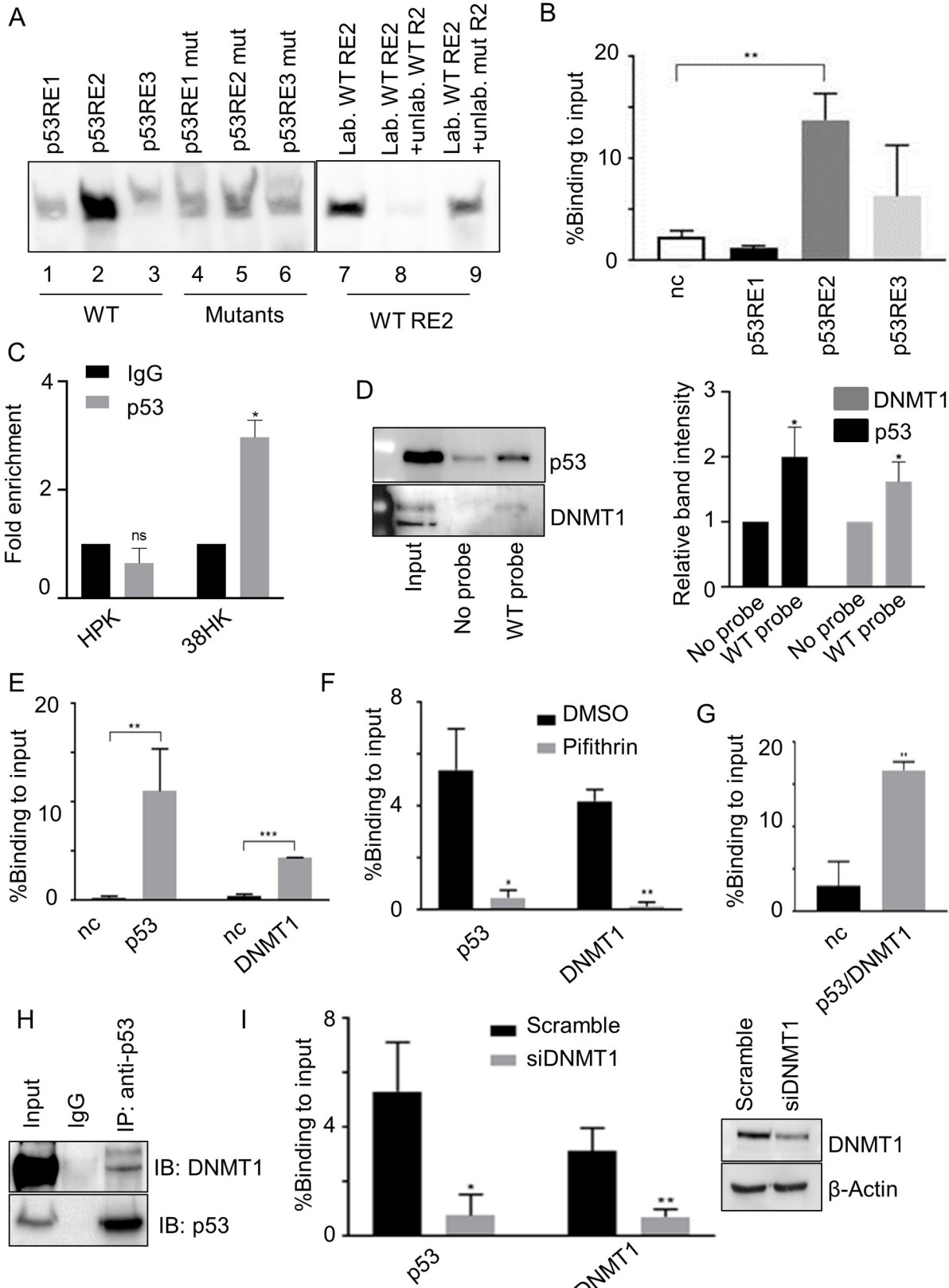

**Fig 2. p53 and DNMT1 form a complex that is recruited to the *ITGA1* promoter.** (A) Electromobility shift assay performed with 38HK nuclear protein extracts and biotinylated probes containing p53RE WT or mutated sequences. Probes were incubated and cross-linked with protein extracts. Unlabeled WT or mutant p53RE2 probes were used as a control. Images shown are representative examples of 2 different

experiments. (B) 38HK were cross-linked and chromatin was processed for ChIP using p53 antibody. Results were analyzed by qPCR with primers spanning p53RE1, p53RE2, p53RE3, or the intergenic region of chromosome 22 as a negative control (nc). Error bars represent standard deviations of 3 independent experiments performed in triplicate. **, $p < 0.01$. (C) HKs or 38HK were cross-linked and chromatin was processed for ChIP using p53 or IgG antibodies. Results were analyzed by qPCR using primers spanning for p53 REs of the *ITGA1* promoter and normalized to IgG enrichment (negative control). Error bars represent the standard deviation of 2 independent experiments performed in 2 different HKs donors. *, $p < 0.05$, ns, not significant. (D) Cell lysate was incubated with WT biotinylated probe containing p53 REs of the *ITGA1* promoter. Incubation without a probe was used as a control. DNA-associated proteins were recovered by precipitation with streptavidin beads and analyzed by IB. Images shown are representative examples of 3 independent experiments. Signals of 3 different IBs were quantified with Image Lab software (right panel). Data shown are the means of 3 independent experiments. *, $p < 0.05$. (E) Chromatin from 38HK was processed for ChIP experiments using p53 or DNMT1 antibodies. Results were obtained by qPCR with primers spanning p53RE2 or the intergenic region of chromosome 22 (nc). Error bars indicate standard deviations from 3 independent experiments performed in duplicate. **, $p < 0.01$; ***, $p < 0.001$. (F) 38HK were cultured in medium containing cyclic pifithrin-α hydrobromide or DMSO as a control. Chromatin was processed for ChIP using p53 or DNMT1 antibodies. Results were obtained by qPCR using primers spanning p53RE2. Data shown are the means of 2 independent experiments performed in triplicate. *, $p < 0.05$, **, $p < 0.01$. (G) Chromatin was processed for a ChIP-reChIP assay in which p53-immunoprecipitated DNA was re-immunoprecipitated by DNMT1. Enrichment of p53RE2 or the intergenic region of chromosome 22 (nc) was obtained by qPCR. Data shown are the means of 3 independent experiments performed in triplicate. **, $p < 0.01$. (H) Nuclear protein extracts from 38HK were processed for IP. Agarose beads were conjugated with IgG or p53 antibodies. Conjugated beads were incubated with protein lysate overnight. IgG was used as a control. Results were obtained by IB using the indicated antibodies. (I) 38HK were transfected with *DNMT1* siRNA or control siRNA (Scramble). After 72 h, a ChIP assay was performed with p53 or DNMT1 antibodies. Results were obtained by qPCR using p53RE2 primers. Error bars represent standard deviations from 3 independent experiments. *, $p < 0.05$; **, $p < 0.01$. DNMT1 protein levels in different cells were determined by IB with the indicated antibodies (right panel).

agent, resulted in activation of *ITGA1* expression (Fig 3G). This event also coincided with the acetylation of histone 3 at K9 (H3K9), which is associated with transcriptional activation (Fig 3H).

Finally, because 38HK also contains high levels of ΔNp73α, which inhibits the expression of p53-regulated genes [25], we evaluated the impact of ΔNp73α depletion on *ITGA1* expression. No significant changes in *ITGA1* mRNA levels were detected in 38HK transfected with ΔNp73α antisense or sense oligonucleotides (S1B Fig). In addition, ΔNp73α does not bind *ITGA1* promoter, as shown by ChIP experiments (S1C Fig).

These findings show that the inhibition of the p53/DNMT1 complex results in activation of *ITGA1* transcription, whereas ΔNp73α does not appear to be involved in the inhibition of *ITGA1* expression.

## Full-length p53 phosphorylated at S392 plays a key role in the inhibition of *ITGA1* expression

As mentioned above, the accumulated p53 in 38HK is phosphorylated only at two serines (15 and 392) [25]. Next, we examined by DNA pulldown assay the binding of p53 S15 and/or S392 phosphorylated forms to the *ITGA1* promoter. The major p53 form, which is able to bind p53RE2 of the *ITGA1* promoter, is phosphorylated at S392 (Fig 4A). It has previously been shown that the double-stranded RNA-dependent protein kinase PKR directly interacts with p53 and phosphorylates S392 [27, 28]. IB showed that HPV38 E6 and E7 activate PKR (Fig 4B). The HPV38-mediated PKR activation was also observed in cancer-derived cells, U2OS, expressing E6 and/or E7 (Fig 4C). We observed that E7 alone is much more efficient than E6 or E6/E7 expressed as polycistronic RNA (Fig 4C). It is possible that this phenomenon is due to a more efficient translation of E7 in comparison to the E6/E7 mRNA. Blocking the activity of PKR by using a chemical inhibitor, 2-aminopurine (2AP), resulted in a small but significant reduction in the levels of the S392 p53 form, indicating that PKR is involved in p53 phosphorylation in 38HK together with other cellular kinase(s) (Fig 4D). Despite the small reduction in S392 p53 protein levels upon treatment with 2AP, *ITGA1* mRNA and protein levels increased considerably in cells exposed to the chemical inhibitor (Fig 4E and 4F). In addition, ChIP assays in 38HK cells treated or not treated with 2AP showed that the recruitment of both p53

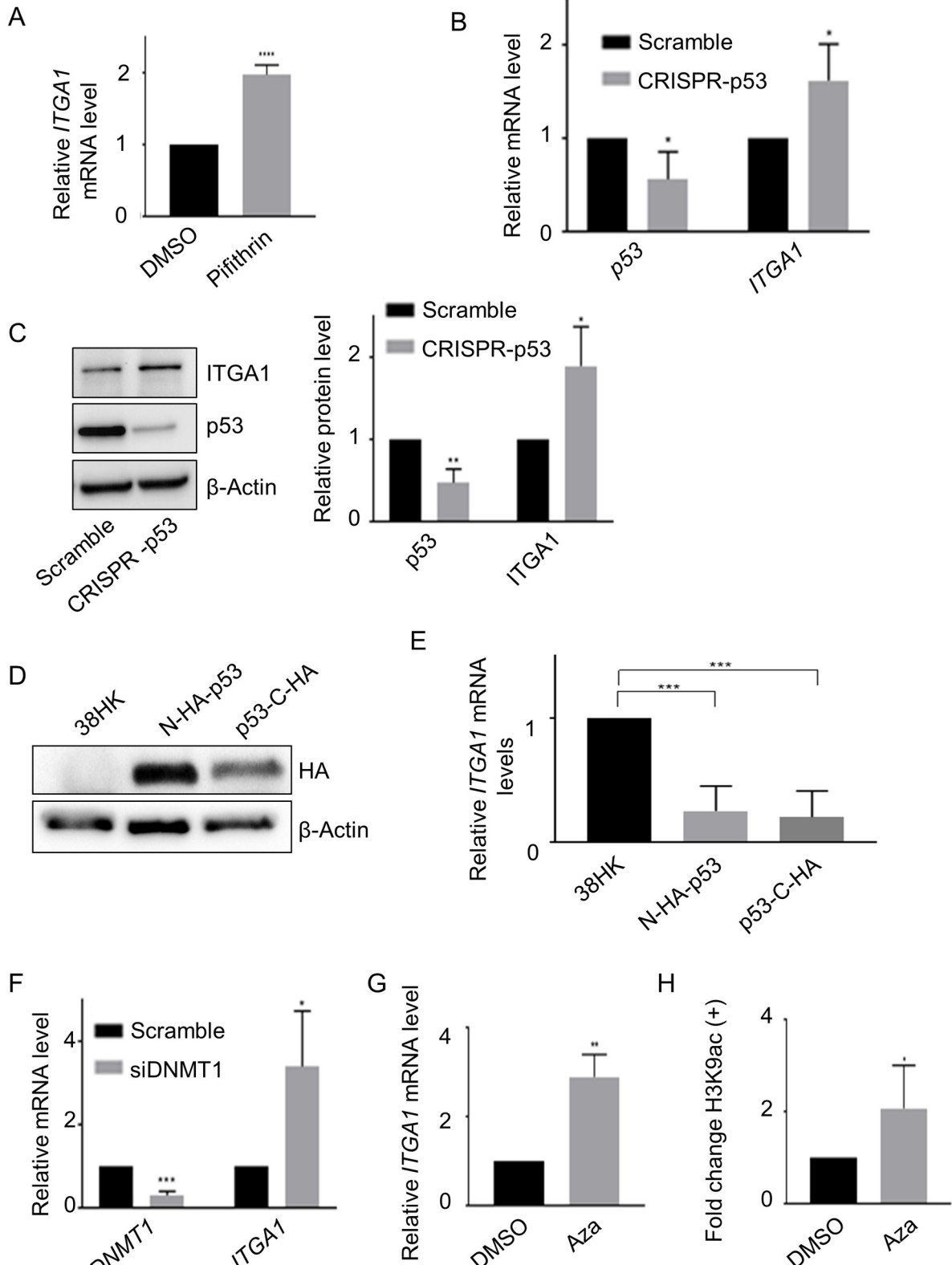

**Fig 3. The p53/DNMT1 complex inhibits *ITGA1* expression.** (A) mRNA levels of 38HK treated with cyclic pifithrin-α hydrobromide or DMSO for 6 h were analyzed by RT-qPCR and normalized to *GAPDH*. Histograms represent the mean of at least 3 independent experiments. ****, $p<0.0001$. (B and C) *ITGA1* and *p53* mRNA and protein levels from 38HK expressing WT p53 (Scramble) or with

CRISPR/Cas9-mediated *p53* deletion (CRISPR-p53) were measured by RT-qPCR (B) and IB (C). (B) Gene expression was normalized to *GAPDH*. (C) Protein quantification was normalized to β-actin. Signals of 4 different IBs were quantified with Image Lab software (right panel). Data shown are the means of 4 independent experiments. *, $p < 0.05$. (D and E) 38HK N-HA-p53 or p53-C-HA cells were generated by retroviral transduction with WT p53 tagged at the N- or C-terminus. As a control, 38HK were transduced with the corresponding empty plasmid. Protein extracts and total mRNA levels were processed for IB and RT-qPCR analysis, respectively. (D) IB images shown are representative examples of 3 independent experiments. (E) *ITGA1* mRNA levels were normalized to *GAPDH*. Error bars indicate standard deviations of 4 independent experiments. ***, $p < 0.001$. (F) 38HK were transfected with control siRNA (Scramble) or with *DNMT1* siRNA (siDNMT1). After 72 h, cells were collected for RNA extraction and RT-qPCR analysis. Error bars indicate standard deviations of 3 independent experiments. *, $p < 0.05$; ***, $p < 0.001$. (G) *ITGA1* expression was evaluated by RT-qPCR after 24 h of treatment with 5-aza-2′-deoxycytidine (Aza) or DMSO at 30 μM final concentration. Error bars represent standard deviations of 3 independent experiments. **, $p < 0.01$. (H) H3K9ac at the *ITGA1* promoter was evaluated by ChIP assay after treatment with Aza or DMSO as previously described ($n = 4$). Results were obtained by qPCR using primers for p53RE2. *, $p < 0.05$.

and DNMT1 cellular proteins was affected by chemical inhibition of PKR phosphorylation (Fig 4G and 4H).

To corroborate these data that indicate cross-talk between p53 and PKR in 38HK, we performed reciprocal IP using PKR or p53 antibodies to assess the possible interaction between the two cellular proteins. The PKR/p53 complex was immunoprecipitated by both antibodies (Fig 5A). Importantly, the S392 p53 form was found to be associated with PKR (Fig 5B). 2AP treatment resulted in a strong decrease in S392 phosphorylation of the PKR-associated p53 form (Fig 5B), whereas no significant changes were observed in the total p53 form co-precipitated with PKR, suggesting that PKR-mediated p53 phosphorylation does not affect the interaction between the two cellular proteins (Fig 5B). A ChIP-reChIP assay using an antibody specific for the T446-phosphorylated PKR form showed that the p53/p446PKR complex is able to bind p53RE2 of the *ITGA1* promoter (Fig 5C). To further characterize the p53/PKR complex, we first fractionated the nuclear extracts of 38HK exposed or not exposed to 2AP by sucrose density gradient ultracentrifugation. Subsequently, the p53 complex was immunoprecipitated in each sucrose gradient fraction (Fig 5D). A trimeric complex containing p53/PKR/DNMT1 was found in some fractions of the sucrose gradient (Fig 5D). In agreement with the data shown in Fig 5B, 2AP treatment did not influence the p53/PKR interaction, whereas DNMT1 was lost from the complex.

Together, these findings provide evidence that the full-length p53 form phosphorylated at S392 by PKR interacts with DNMT1 and inhibits *ITGA1* expression.

## Ectopic overexpression of *ITGA1* or p53 has different impacts on 38HK proliferation

Next, to understand the biological significance of *ITGA1* downregulation in 38HK, we investigated the impact of *ITGA1* overexpression on 38HK proliferation using a colony formation assay. Cells were transfected with a construct expressing *ITGA1* and the zeocin resistance gene and cultured under antibiotic selection. We observed a significant decrease in colony formation in 38HK expressing ectopic levels of *ITGA1* compared with mock-treated cells (Fig 6A). In addition, analysis of the cell-cycle profile by flow cytometry showed that *ITGA1* overexpression significantly increased the sub-G0 cell population, which is a sign of cellular death (Fig 6B).

It has previously been reported that ITGA1 negatively regulates EGFR signaling by promoting EGFR de-phosphorylation, with consequent inhibition of cellular proliferation [23]. Therefore, we evaluated the status of EGFR signaling in 38HK after *ITGA1* ectopic expression by determining the levels of cyclin D1 (CCND1), which is positively regulated by activation of EGFR signaling [29, 30]. In accordance with the inhibition of cellular proliferation, CCND1 protein levels decreased upon ectopic expression of *ITGA1* (Fig 6C). Surprisingly, we also observed a reduction in EGFR protein levels upon *ITGA1* overexpression. However, no

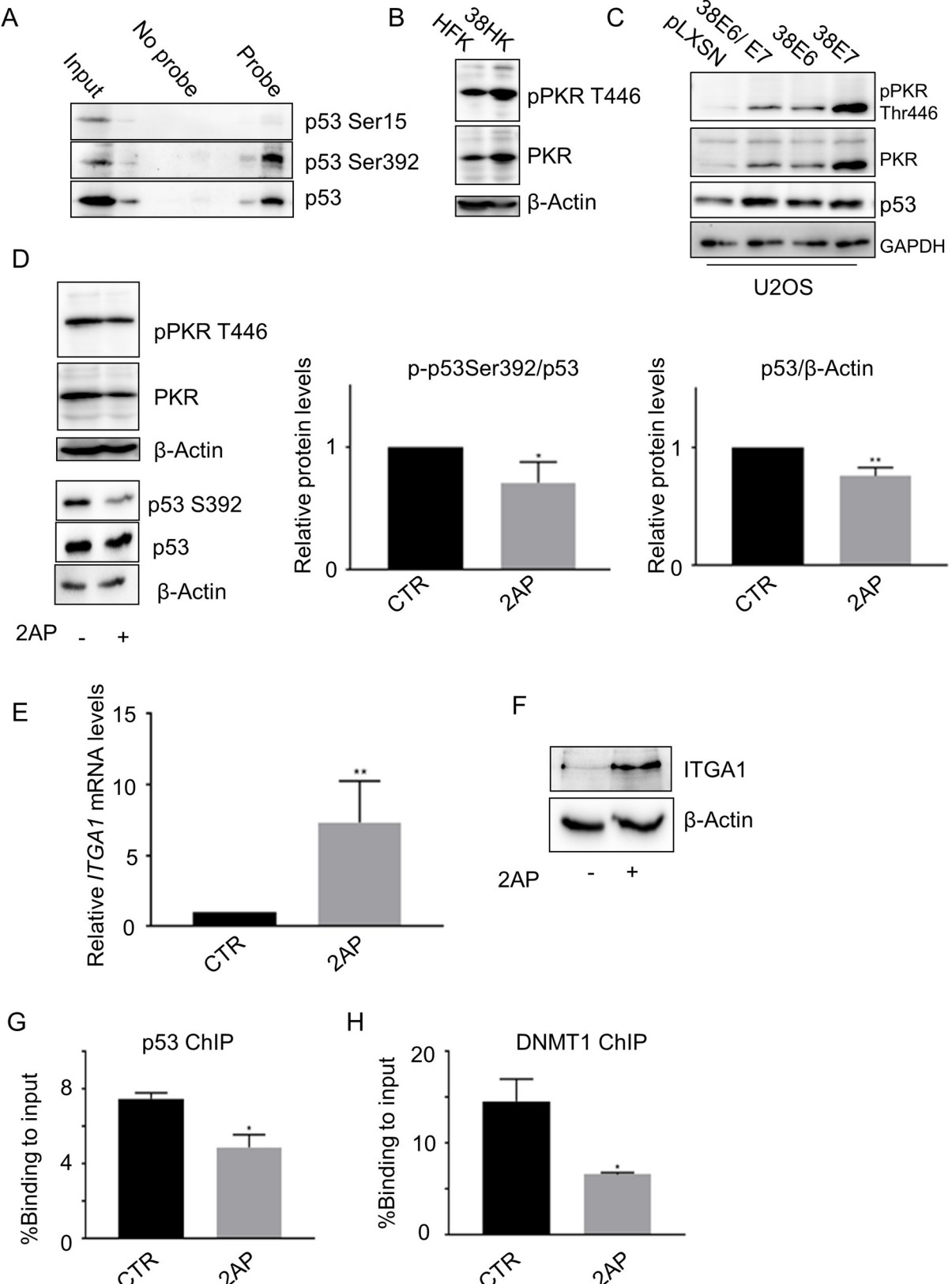

**Fig 4. Full-length p53 phosphorylated at S392 plays a key role in *ITGA1* inhibition.** (A) Protein extracts from 38HK were processed for oligonucleotide pulldown as previously described. Images shown are representative examples of 3 independent experiments. (B) HK and

38HK were processed for protein extraction and IB with the indicated antibodies. After incubation with p446PKR antibody, the membrane was stripped and incubated with total PKR antibody. (C) Proteins extracts from U2OS cells retrovirally transduced with E6 and/or E7 from HPV38 were analyzed by IB with the indicated antibodies. (D) 38HK were treated with PKR inhibitor, 2AP, or PBS:glacial acetic acid (200:1) as a control for 4 h at 10 mM final concentration. p-p53 S392 and p53 band intensities were quantified and normalized to total p53 (central panel) or β-actin (right panel). Membranes were first incubated with p446PKR, then stripped and incubated with total PKR. Data shown are the means of 3 independent experiments. *, $p < 0.05$, **, $p < 0.01$. (E and F) 38HK were treated with 2AP, and *ITGA1* mRNA (D) and protein levels (E) were determined by RT-PCR and IB, respectively. (D) Data shown are the means of 3 independent experiments (**, $p < 0.01$). (E) Images shown are representative examples of 3 independent experiments. (G and H) ChIP assay using p53 or DNMT1 antibodies was performed in 38HK treated with or PBS:glacial acetic acid (200:1) as a control (CTR) or 2AP. Data shown are the means of 2 independent experiments performed in duplicate by qPCR using p53RE2 primers. *, $p < 0.05$.

significant changes in *EGFR* mRNA levels were observed upon *ITGA1* overexpression in 38HK and mock-treated cells (Fig 6D), suggesting that EGFR destabilization in the presence of ITGA1 is mediated post-translationally.

To corroborate the link between p53-mediated *ITGA1* downregulation and cellular proliferation, we evaluated whether loss of the *p53* gene could influence 38HK growth. *TP53* deletion by CRISPR/Cas9 inhibited cellular proliferation (Fig 7A and 7B). Similar results were obtained by inhibiting p53 functions by using pifithrin (S1D Fig). Next, we evaluated whether the decrease in 38HK proliferation in CRISPR-p53 38HK is dependent on the loss of the full-length p53 form, and not on the loss of the truncated p53 isoforms. We generated a retroviral vector that expresses an N-terminus HA-tagged *p53* gene (Δ-CRISPR), in which the third base of several codons was mutated (Fig 7C), without altering the amino acid sequence. Thus, this mutated *p53* gene encodes a WT protein but, due to the alteration of DNA sequence, it is not targeted by the guide RNA, which was designed to delete the endogenous *p53* gene. In addition, two LoxP elements were located immediately upstream and downstream of the Δ-CRISPR *p53* gene in order to modulate its expression via Cre recombinase. The Cre recombinase gene fused to a triple-mutant form of the human estrogen receptor that gains access to the nuclear compartment only after exposure to 4-hydroxytamoxifen (TMX) but not to the natural ligand 17β-estradiol was cloned in a second retroviral vector (Fig 7C). 38HK were sequentially transduced with two recombinant retroviruses, and subsequently the endogenous *p53* gene was deleted by CRISPR/Cas9. We observed that the modified 38HK line expressing ectopic levels of the Δ-CRISPR *p53* gene had a higher proliferation rate compared with 38HK (Fig 7D). Importantly, after addition of TMX and loss of p53, the proliferation of these cells was rapidly reduced, whereas no significant effect was observed when TMX was added to 38HK (Fig 7D and 7E). To corroborate these findings, we transduced 38HK with recombinant retroviruses that enable the synthesis of full length p53 fused to the HA-tag at the N- or C-terminus and retained the ability to repress ITGA1 expression (Fig 3D and 3E). Both HA-p53 fusion proteins were able to stimulate the proliferation of 38HK (Figs 7F and 7G and S1E).

These findings show that p53 exerts pro-proliferation functions in 38HK via inhibition of ITGA1 expression and EGFR signaling activation.

## HPV38 E6 and E7 expression in the skin of transgenic mice inhibits ITGA1 transcription

To corroborate our findings in *in vitro* experimental models, we investigated whether HPV38 E6 and E7 expression has the ability to alter *Itga1* expression in skin keratinocytes in mice. We previously developed a transgenic mouse model that expresses HPV38 E6 and E7 in the keratinocytes of the skin basal layer under the control of keratin 14 promoter (K14) [31]. After isolating skin keratinocytes from WT and HPV38 E6/E7 transgenic mice, we determined *Itga1* mRNA levels by quantitative RT-PCR. The viral proteins inhibited *Itga1* expression in skin keratinocytes (Fig 8A), confirming the *in vitro* findings.

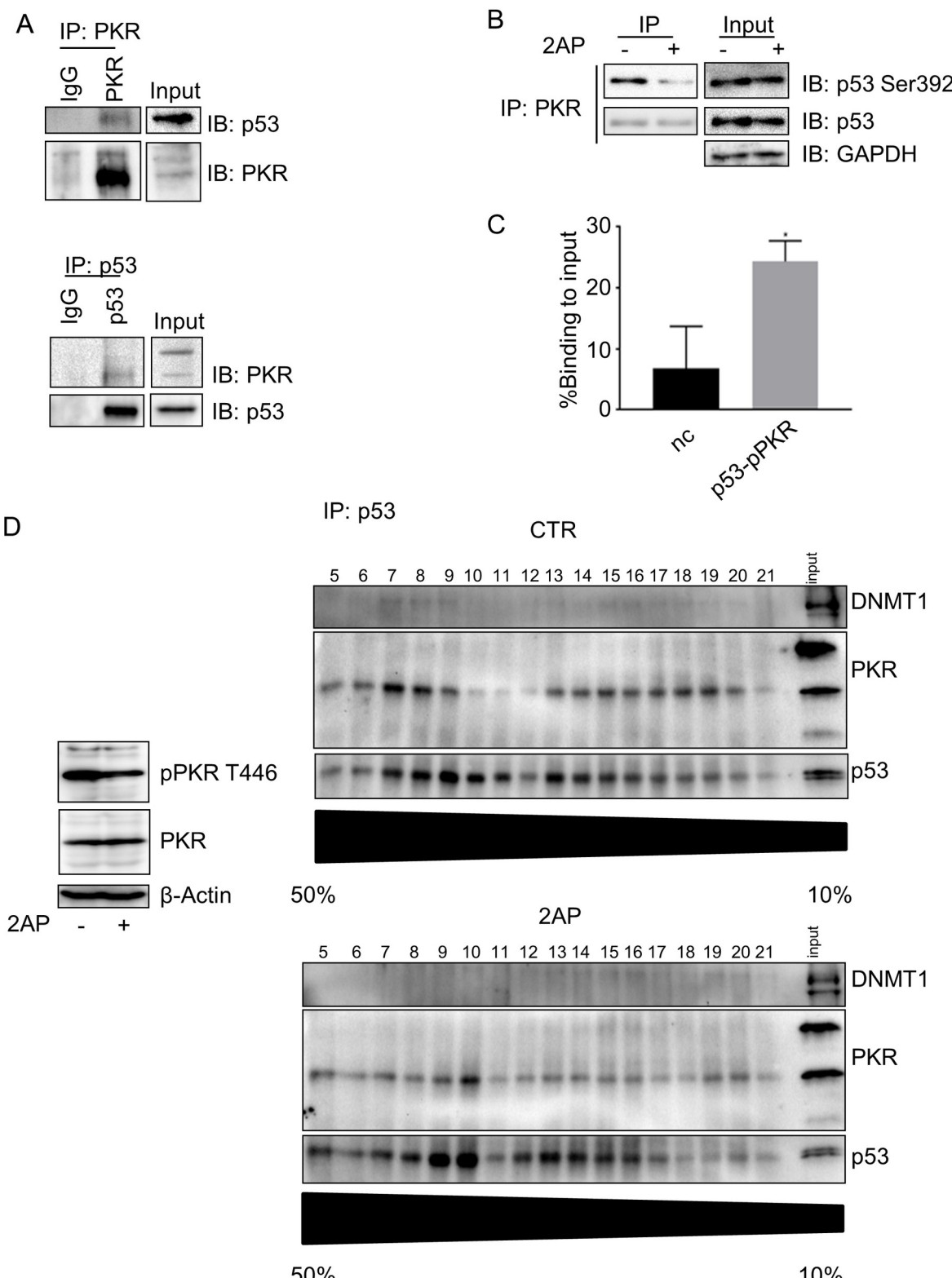

**Fig 5. PKR inhibition reduces p53 phosphorylation and regulates *ITGA1* expression.** (A) Protein extracts from 38HK were processed for IP. Agarose beads were conjugated with total PKR antibody (top) or p53 antibody (bottom). Conjugated beads were incubated with protein lysate overnight. IgG was used as a control. Results were obtained by IB. Images were cropped due to the presence of samples irrelevant to this work. Images shown are representative examples of 2 independent experiments. (B) Total protein extracts from cells

treated with PKR inhibitor, 2AP, or PBS:glacial acetic acid (200:1) as a mock-treated control (-) were obtained for PKR IP as previously described (top). The input was run in a different gel to improve image quality. (C) 38HK were cross-linked and chromatin was extracted for a ChIP-reChIP assay by p53 IP followed by p446PKR IP. Data from 3 independent experiments performed in duplicate were analyzed by qPCR. p53RE2 and the intergenic region of chromosome 22 (nc) primers were used. *, $p < 0.05$. (D) 38HK were treated with PKR inhibitor, 2AP, or PBS:glacial acetic acid (200:1)as a control for 4 h at 10 mM final concentration. Nuclear extracts were used for 50% to 10% sucrose gradient protein complex isolation. Fractions obtained were immunoprecipitated with p53 antibody. Results were analyzed by IB.

We have recently shown that HPV38 E6/E7 transgenic mice are highly susceptible to UV-induced DNA mutations and cSCC development compared with WT animals [32]. By whole-exome sequencing, we observed that these animals, upon long-term UV exposure, accumulate mutations in crucial cancer-linked genes, including *p53* [32]. Therefore, we next determined whether *p53* mutations in the core DBD detected in 3 different cSCC may result in loss of inhibition of *Itga1* expression. Quantitative RT-PCR experiments showed that *p53* mutations correlated with an increase in *Itga1* expression in 2 cSCC (Fig 8A). However, we determined that in 2 cSCC, the *Itga1* gene contained deleterious non-synonymous mutations (Fig 8B). Although the third cSCC expressed high levels of WT *Itga1*, it contained a non-synonymous, but not deleterious, mutation in a region of the *Egfr* gene encoding the tyrosine kinase domain (Fig 8B). Mutations in this EGFR domain have been identified in human cancer and result in activation of EGFR signaling [33–35].

In summary, these results confirm the ability of HPV38 to inhibit *ITGA1* expression and highlight the importance of *ITGA1* inactivation in UV-induced cSCC development.

## Discussion

Inactivation of p53 is a key step in cellular transformation. In approximately 50% of human cancers, p53 is inactivated by DNA mutation, which frequently occurs in the DBD. As a consequence, mutated p53 loses its normal transcriptional functions as a tumor suppressor [36]. Importantly, a vast number of studies have shown that p53 mutations, in addition to the disruption of the tumor suppressor function of p53, can also confer oncogenic gain-of-function activities [11, 12]. Several findings support the model that gain-of-function p53 mutations induce conformational changes that allow mutated p53 to interact with other cellular proteins, including products of tumor suppressor genes or oncogenes as well as specific promoter REs [36]. Cellular response to a broad spectrum of stresses leads to post-translational modification of WT p53, which can be phosphorylated, acetylated, and ubiquitinated at specific serine, threonine, and lysine residues, respectively [1, 2]. Similarly, mutated p53 forms are post-translationally modified at specific residues, with consequent acquisition of more aggressive oncogenic functions. For instance, it has been shown that S392 is one of the most frequently phosphorylated residues in the mutated p53 forms [37, 38]. Although the *p53* gene is highly mutated, approximately 50% of human cancers retain the WT *p53* gene, and its tumor suppressor functions can be altered by additional mechanisms, for example by overexpression of truncated N-terminal isoforms of p53 and p73 that act as dominant-negative mutants of p53 [39]. Other plausible models of alteration of WT p53 tumor suppression functions can rely on specific patterns of post-translational modifications and interactions with cellular proteins. In this study, we describe that expression of HPV38 E6 and E7 in HKs promotes the formation of a transcriptionally repressive S392 phosphorylated p53/DNMT1 complex on the *ITGA1* promoter. Inhibition of ITGA1 expression results in activation of EGFR signaling and cellular proliferation. Paradoxically, ectopic levels of WT p53 in 38HK further repressed ITGA1 expression and increased cellular proliferation. Thus, in HPV38-transformed HKs, WT p53 acquires pro-proliferation properties. This conclusion is further corroborated by the fact that

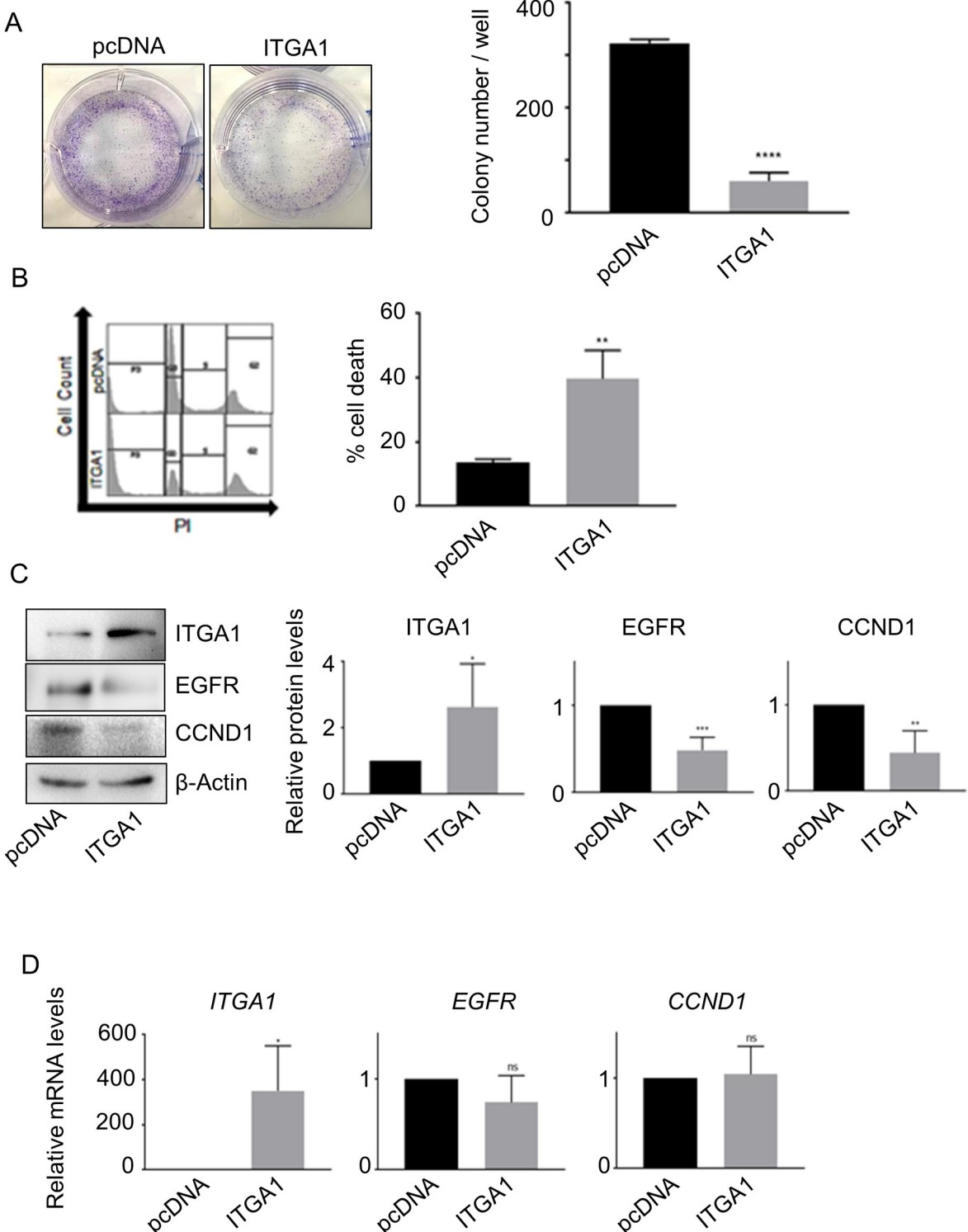

**Fig 6. Alteration of the ITGA1-regulated network is implicated in cellular proliferation and transformation.** (A) 38HK were transfected with *ITGA1* cDNA (ITGA1) or empty plasmid control (pcDNA). After zeocin selection, 38HK were fixed with crystal violet (left panel) and total colony number per well was counted (right panel). Data shown are the means of 3 independent experiments. ****, $p < 0.0001$. (B) 38HK transfected with *ITGA1* cDNA (ITGA1) or empty plasmid control (pcDNA) were fixed and stained with propidium iodide for flow cytometry analysis. The histograms (right) represent the means of the sub-G0 population of 3 independent experiments. **, $p < 0.01$. (C and D) Total protein and mRNA extracts from transfected 38HK were analyzed by IB and RT-qPCR. (C) Protein band

intensity was quantified and normalized to β-actin. (D) *ITGA1*, *EGFR*, and *CCND1* mRNA levels were normalized to *GAPDH*. For (C) and (D), data shown are the means of 4 independent experiments *, $p<0.05$; **, $p<0.01$; ***, $p<0.001$; ns, not significant.

38HK are addicted to WT p53 for their cellular proliferation. Deletion of the *p53* gene by CRISPR/Cas9 strongly inhibited cellular growth. Interestingly, a previous study described the formation of a p53/DNMT1 complex with a transcriptional repressive function in a non-virus-related experimental model [40]. Therefore, it is plausible to hypothesize that HPV38 oncoproteins exploit cellular mechanisms that can be generated in other contexts. Our data show that p53 phosphorylation at S392 by PKR is essential for the interaction of p53 with DNMT1. Although PKR was initially considered to be a tumor suppressor, it is now well demonstrated that it also exerts oncogenic functions, being overexpressed and activated in many types of cancers, including several hematopoietic malignancies [41]. Interestingly, similar observations have been made for the DNMT protein family (reviewed in [42]). Based on these findings, we could hypothesize that HPV38 E6 and E7 generate a specific scenario in the infected cells, in which PKR and DNMT1 act as an oncoprotein. The described link between HPV38, p53 and ITGA1 appears to be not shared with the mucosal high-risk HPV types that induce p53 degradation. Indeed, ITGA1 is more expressed in HPV16-positive than HPV16-negative oropharyngeal cancers [43].

The link of p53 to the integrin network has already been described in several independent studies; however, the mechanisms involved in these events have been poorly characterized [44]. Our findings describe a novel p53 mechanism in the regulation of the integrin network and stimulation of cellular proliferation, by forming a transcriptional repressive complex that is recruited to the *ITGA1* promoter. In support of our model, it has been shown in an independent study by ChIP-seq experiments that in proliferative primary human fibroblasts, p53 binds the same p53RE2 of the *ITGA1* promoter as observed in 38HK [45]. Importantly, senescence of the fibroblasts resulted in loss of p53 from the *ITGA1* promoter, indicating that the recruitment of p53 to the *ITGA1* promoter is associated with proliferative events [45].

Our experiments in transgenic animals also support a key role of ITGA1 inactivation in cellular transformation. We showed that HPV38 E6/E7 expression in the skin of transgenic mice strongly inhibits *Itga1* expression. Many functional studies support the model that beta HPV types, via the interaction of the viral proteins E6 and E7, facilitate the accumulation of UV-induced DNA mutations, increasing the susceptibility to skin carcinogenesis [17]. However, the viral proteins appear to be dispensable after full development of cSCC. Our findings indicate that in the HPV38 E6/E7 transgenic animals exposed to long-term UV irradiation, ITGA1 inactivation must be constantly present during all steps of skin carcinogenesis. *Itga1* expression is inhibited at early stages by the viral oncoproteins, whereas at later stages the gene is inactivated by DNA mutations. These findings further support the concept that beta HPV types act with a "hit-and-run" mechanism in promoting cSCC development.

In conclusion, here we described a novel virus-mediated mechanism that converts WT p53 into an oncoprotein. This WT p53 form acquires the properties to interact with PKR and DNMT1 and to repress cellular gene expression. It will be important to evaluate whether similar mechanisms occur in cancer cells of different origins, offering the possibility to develop novel anti-cancer therapeutic strategies.

## Materials and methods

### Cell cultures and treatments

The experiments were carried out in HKs isolated from neonatal foreskin and in a HK cell line expressing the *hTERT* gene, in order to prolong the life span of the cells. HKs, hTERT HKs

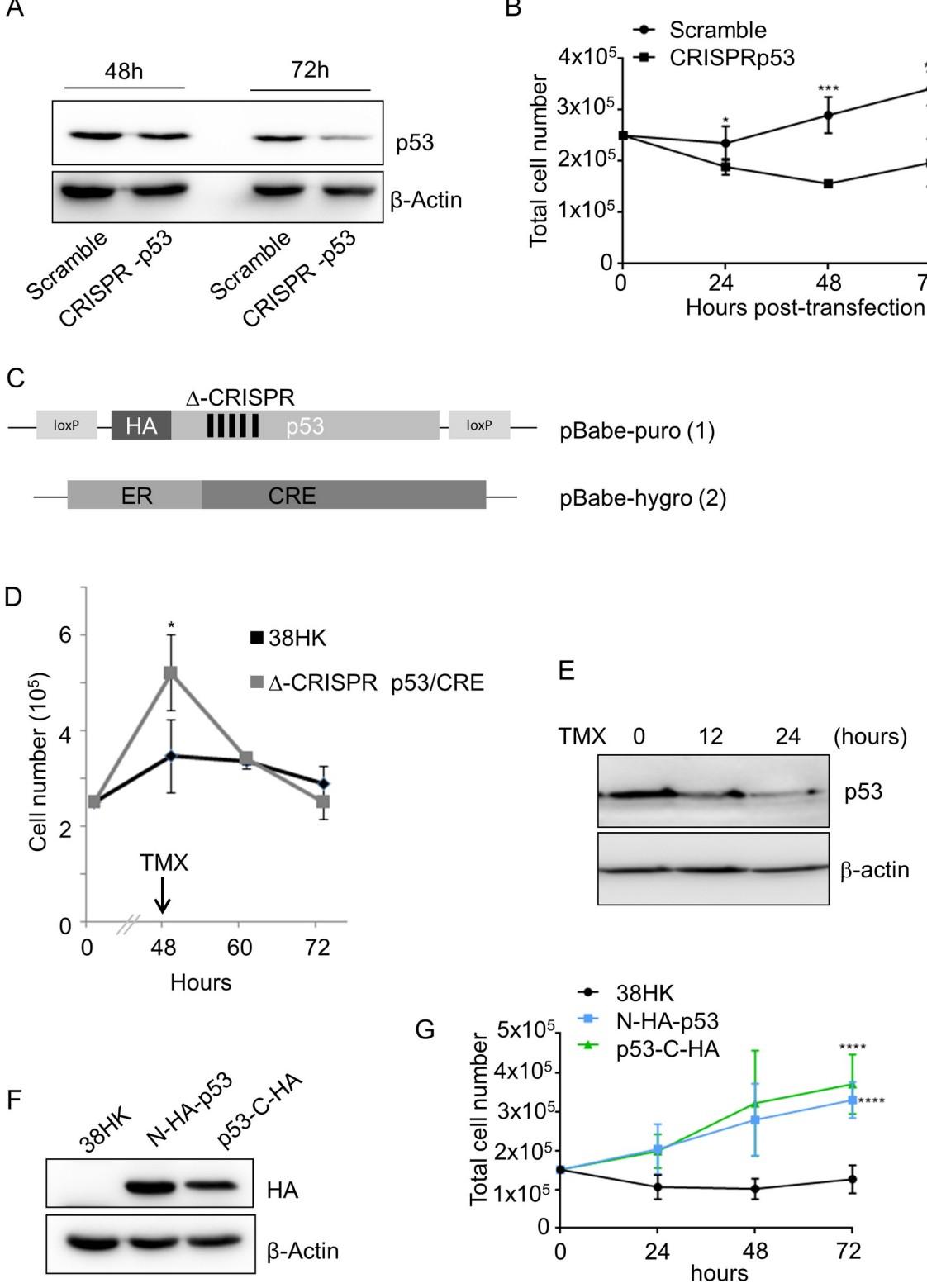

**Fig 7. Full-length WT p53 displays pro-proliferation properties in 38HK.** (A and B) 38HK were seeded into 6-well plates and transfected with Scramble or CRISPR-p53 plasmid for *p53* knockdown. (A) Protein extracts were processed for IB 48 h and 72 h after transfection. Images shown are representative examples of 2 independent experiments. (B) Cells were also collected at 24, 48, and 72 h after transfection, stained with trypan blue, and counted. Error bars represent the standard deviations of 2 independent experiments

performed in duplicate. *, $p<0.05$, **, $p<0.01$; ***, $p<0.001$. (C) Schematic representation of retroviral plasmids containing WT p53 HA-tagged and flanked by LoxP sites (1) or Cre recombinase fused to a triple-mutant form of the human estrogen receptor (ER) (2). In addition, the *p53* codon sequence was modified to avoid CRISPR/Cas9 targeting (1). (D and E) 38HK were transduced with the plasmids previously described (Δ-CRISPR p53/CRE) and transfected with CRISPR/Cas9 p53 plasmids to knock out endogenous p53. $2.5 \times 10^5$ cells were seeded into 6-well plates and allowed to grow for 48 h. TMX was added to the medium at 2 μM final concentration. (D) Cell number was quantified by trypan blue staining at 12 h and 24 h after treatment. Data shown are the means of 3 independent experiments. *, $p<0.05$. (E) IB shows p53 protein levels after TMX treatment. (F) 38HK N-HA-p53 or p53-C-HA cells were generated by retroviral transduction with WT p53 tagged at the N- or C-terminus. As a control, 38HK were transduced with the corresponding empty plasmid. Protein extracts were processed for IB. Images shown are representative examples of 3 independent experiments. (G) 38HK N-HA-p53 or p53-C-HA cells were also seeded into 6-well plates. After 24, 48, and 72 h, cells were collected, stained with trypan blue, and counted. Data shown are the means of 3 independent experiments performed in duplicate. ****, $p<0.0001$.

and U2OS stably expressing HPV38 E6 and/or E7 as well as p53HA-Tag 38HK were generated by retroviral transduction. A Δ-CRISPR p53/CRE cell line was obtained after sequential transduction with a retroviral vector that expresses a N-terminus HA-tagged *p53* gene (Δ-CRISPR) flanked by LoxP elements and a retroviral vector containing Cre recombinase fused to the human estrogen receptor. The third base of several codons of the *p53* gene was mutated to avoid RNA targeting that was designed to delete the endogenous *p53* gene [21]. The 38HK, p53HA-Tag 38HK, hTERT HKs, and Δ-CRISPR p53/CRE cell lines were cultured together with NIH 3T3 feeder layers in FAD medium containing 3 parts Ham's F12, 1 part DMEM, 2.5% FCS, insulin (5 μg/mL), epidermal growth factor (10 ng/mL), cholera toxin (8.4 ng/mL), adenine (24 μg/mL), hydrocortisone (0.4 μg/mL), and 1% of penicillin/streptomycin preparation. Feeder layers were prepared by treating NIH 3T3 cells with mitomycin for 2 h. NIH 3T3 cells and U2OS were cultured in DMEM supplemented with 10% FCS and 1% penicillin/streptomycin preparation.

Transient transfection experiments were performed using Lipofectamine 2000 transfection reagent (Invitrogen) or TransIT-Keratinocytes Transfection Reagent (Mirus) according to the manufacturer's protocols.

Cells were incubated for 6 h in medium containing cyclic pifithrin-α hydrobromide at 20 μM (Sigma); others were incubated for 24 h in medium containing 5-aza-2′-deoxycytidine at 30 μM (Sigma).

2AP (Sigma) was prepared in PBS:glacial acetic acid (200:1). Cells were treated for 4 h at 10 mM final concentration; PBS:glacial acetic acid (200:1) was used as a mock-treated control.

TMX (Sigma) was reconstituted in ethanol. Cells were treated for 12 h or 24 h at 2 μM final concentration.

For FACS staining, cells were collected, washed twice in PBS, and fixed in 70% of ethanol for 30 min in ice. Samples were stained with propidium iodide at 5 μg/mL final concentration. Subsequently, cells were analyzed with the FACSCanto system (Becton Dickinson).

For the colony formation assay, cells were transfected using TransIT-Keratinocyte Transfection Reagent (Mirus) according to the manufacturer's protocols. Cells were transfected with pcDNA 3.1/Zeo (mock) or pcDNA 3.1/Zeo expressing *ITGA1* (2.5 μg) (a gift from A. Pozzi, Vanderbilt University). After 24 h, the cells were split for selection in zeocin (InvivoGen). They were diluted 10, 100, and 1000 times and were allowed to grow for 4 days. After this period, the colonies were fixed and stained as described previously [46].

For determination of cell growth curves, $1.5-2.5 \times 10^5$ cells were seeded into 6-well plates. After 24, 48, and 72 h, cells were collected and stained with trypan blue (1:1) (Bio-Rad). Samples were counted in duplicate with the TC20 automated cell counter (Bio-Rad).

Cell viability was determined by the MTS assay. Briefly, 20 μL of CellTiter 96 AQueous One Solution Cell Proliferation Assay (Promega) was added to $1.5 \times 10^4$ cells in 96-well plates and

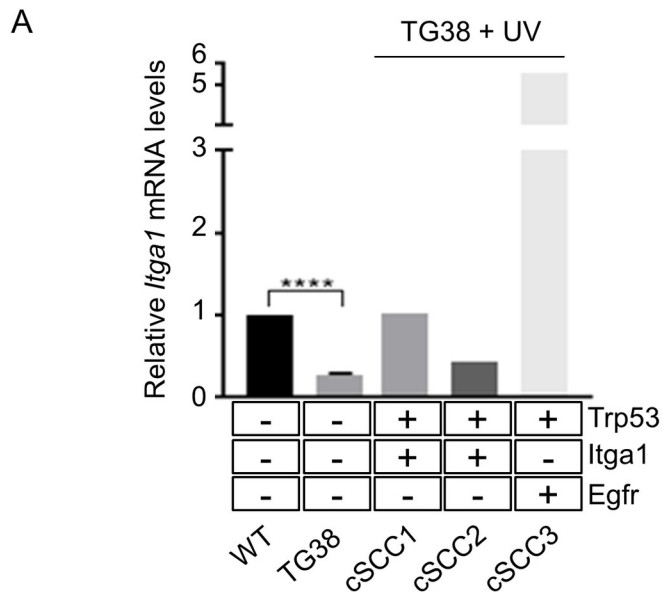

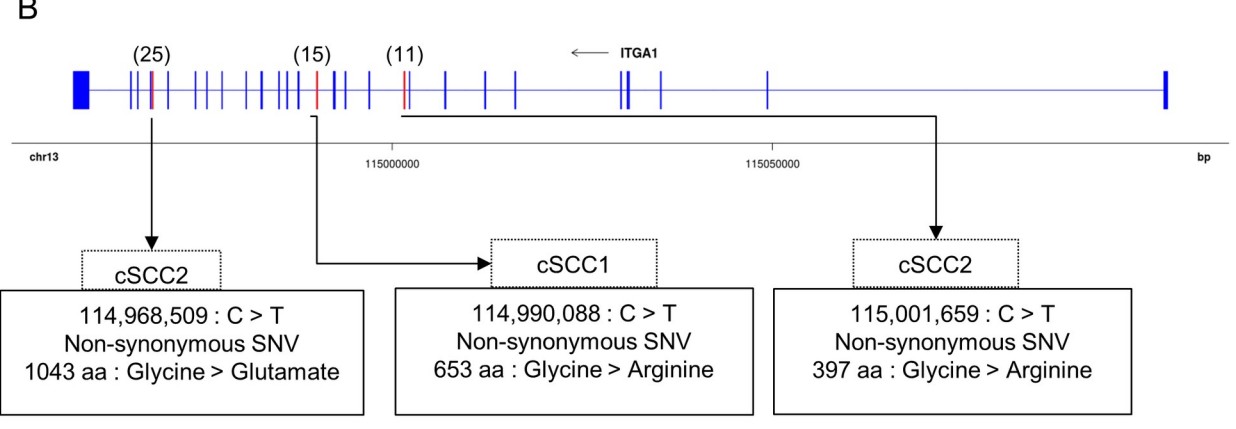

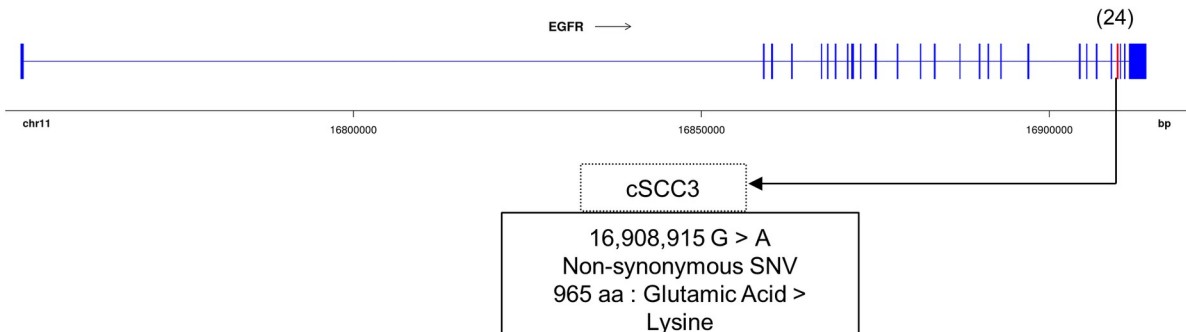

**Fig 8. Itga1 is downregulated and mutated in transgenic mice expressing HPV38 E6/E7.** (A) Skin keratinocytes were isolated from WT animals (*n* = 4) and K14 HPV38 E6/E7 transgenic mice (*n* = 3). After 30 weeks of UV irradiation, cSCC samples (cSCC1–3) were isolated from HPV38 E/E7 transgenic mice. Total RNA extraction was performed and *Itga1* mRNA levels were determined by quantitative RT-PCR by normalizing to *Gapdh*. Whole-exome sequencing of the same mice was also performed. Mutational analysis of the *Trp53*, *Itga1*, and *Egfr* genes was performed as described in Materials and Methods. (B) Genomic position of the exonic mutations and the corresponding amino acid change

are represented for the *Itga1* and *Egfr* genes. WT exons are represented as blue boxes, and mutated exons are represented in red boxes. Text boxes describe the cSCC sample, the genomic position of the nucleotide change on the GRCm38/mm10 mouse reference genome, the type of mutation, and the corresponding amino acid change.

incubated at 37°C for 24, 48, and 72 h. Absorbance at 490 nm was read with the Multiskan GO spectrophotometer (Thermo Scientific) in duplicate. The blank absorbance was subtracted.

### Transgenic mice

DNA and mRNA were extracted from normal and cancer mouse tissues as previously described [31]. A detailed description of the HPV38 E6/E7 transgenic mouse line can be found here: https://mito.dkfz.de/mito/Animal%20line/10954. A detailed description of the UV-induced skin carcinogenesis protocol can be found here: https://mito.dkfz.de/mito/Tumor%20model/10474.

### Ethics statement

The animal facility of the German Cancer Research Center was officially approved by the responsible authority (Regional Council of Karlsruhe, Schlossplatz 4–6, 76131 Karlsruhe, Germany; official approval file number 35–9185.64). The housing conditions are thus in accordance with the German Animal Welfare Act (TierSchG) and EU Directive 425 2010/63/EU. Regular inspections of the facility are conducted by the Veterinary Authority of Heidelberg (Bergheimer Str. 69, 69115 Heidelberg, Germany). All experiments were in accordance with the institutional guidelines (designated veterinarian according to article 25 of Directive 2010/63/EU and animal-welfare body according to article 27 of Directive 2010/63/EU) and were officially approved by the Regional Council of Karlsruhe (file number 35–9185.81/G-64/13 and 35–9185.81/G-200/15).

### Gene silencing

Gene silencing of *DNMT1* was achieved using synthetic siRNA (S1 Table). siRNA or scrambled RNA at 250 nM was transfected using TransIT-Keratinocyte Transfection Reagent (Mirus) according to the standard protocol. Cells were collected after 72 h.

Plasmids for CRISPR/Cas9 were obtained from the Addgene plasmid repository. All single-guide RNAs were designed by Thermo Fisher Scientific. The target sequence information is shown in S1 Table. The CRISPR/Cas9 vectors were generated according to the manufacturer's protocols and then transiently transfected into HKs. Purification of the cells carrying the CRISPR/Cas9 vectors was performed 48 h after transfection according to the manufacturer's protocol (GeneArt CRISPR Nuclease Vector Kit; Life Technologies).

### Reverse transcription and quantitative PCR

For the experiments in *in vitro* models, total RNA was extracted using the NucleoSpin RNA II Kit (Macherey Nagel). The RNA obtained was reverse-transcribed to cDNA using the RevertAid H Minus First Strand cDNA Synthesis Kit (Life Technologies) according to the manufacturer's protocols. Real-time quantitative PCR (qPCR) was performed using the Mesa Green qPCR MasterMix Plus for SYBR Assay (Eurogentec) with the primers listed in S2 Table.

For the experiments in mice, total RNA was isolated from dorsal skin of WT (*n* = 4), K14 HPV38 E6/E7 transgenic animals (*n* = 3), histologically confirmed pre-malignant lesions (actinic keratosis), and cSCC from 3 independent mice. cDNA was synthesized from 1 µg of total RNA using M-MLV reverse transcriptase (Invitrogen), and a mix of random hexamers

were used as primers. RT-qPCR was performed using LightCycler 480 SYBR Green I Master (Roche) with specific mouse primers (S2 Table).

A TaqMan assay was performed with the *ITGA1* TaqMan gene expression assay probe (Hs00235006_m1; Life Technologies), following the manufacturer's instructions. Reactions were run in triplicate, and expression was normalized to *GAPDH* (Hs99999905_m1; Thermo Fisher).

### Fluorescence microscopy

hTERT pLXSN or hTERT HPV38 E6/E7 cells were plated onto coverslips and cultured for 24 h. Thereafter, the cells were fixed with 4% formaldehyde for 30 min at room temperature and permeabilized with 0.1% Triton X. Cells were incubated with blocking solution for 1 h, washed with PBS, and incubated with ITGA1 (ab243033; Abcam) antibody overnight at 4˚C. The cells were then incubated with Alexa Fluor 488-conjugated secondary antibody for 2 h at room temperature and mounted on slides using Vectashield Antifade Mounting Medium with DAPI. The slides were visualized using a Nikon Eclipse Ti wide-field inverted fluorescence video microscope. The images thus captured were analyzed by NIS-Element software from Nikon and processed using ImageJ software.

### Immunoblotting

Cells were lysed using IP buffer (20 mM Tris HCl [pH 7.5], 200 mM NaCl, 1 mM EDTA, 0.5% NP-40) supplemented with Complete Protease Inhibitor mixture (Roche). Samples were resolved by SDS–PAGE and transferred to polyvinylidene difluoride membranes (Perkin Elmer). Membranes were blocked in 5% non-fat milk and incubated overnight at 4˚C with the appropriate primary antibody. Membranes were probed with the following primary antibodies: β-actin (clone C4; MP Biomedicals), GAPDH (6C5) (sc-32233; Santa Cruz), p53 (DO-1) (sc-126; Santa Cruz Biotechnology), DNMT1 (clone 60B1220.1; MAB0079; Abnova), PKR (3072; Cell Signaling Technology), phosphorylated PKR Thr 446 (PA5-37704; Thermo Fischer Scientific), EGFR (4267; Cell Signaling Technology), CCND1 (2978; Cell Signaling Technology), HA-tag (3F10; Roche), and ITGA1 (106267; Abcam).

Images were produced using the ChemiDoc XRS imaging system (Bio-Rad).

### Chromatin immunoprecipitation

ChIP was performed using the Shearing ChIP and OneDay ChIP kits (Diagenode) according to the manufacturer's instructions. Briefly, cells were sonicated to obtain DNA fragments of 200–500 bp. Sheared chromatin was immunoprecipitated the indicated antibodies: p53 (DO-1) (sc-126; Santa Cruz Biotechnology), DNMT1 (clone 60B1220.1; Abnova) phospho PKR T446 (ab32036), p73 (ab215038) and isotype IgG control (Diagenode). For ChIP-reChIP experiments, bead-bound protein–DNA complexes obtained after the first ChIP were incubated with 10 mM Reverse dithiothreitol (DTT) for 30 min at 37˚C with shaking at 400 rpm. Supernatant was collected after centrifugation at 12,000*g* for 1 min. Pelleted beads were incubated again with 10 mM DTT for 20 min at 37ºC and centrifuged at 12,000*g* for 1 min. Then, 10% of the combined supernatants were kept as the input for the second ChIP, which was performed with the OneDay ChIP kit (Diagenode) according to the manufacturer's protocol.

For histone ChIP, the chromatin shearing kit Low SDS and Auto iDeal ChIP-seq kit for Histones (Diagenode) were used together with the SX-8G IP-Star Compact Automated System and histone H3K9ac antibody (Euromedex).

The eluted DNA was used as a template for qPCR. The negative control region chromosome 22 intergenic region as previously described [47] (S2 Table). A negative control region previously described lacking p73 and p53 binding sites was used for ΔNp73α ChIP [48][49].

## Oligonucleotide pulldown assay

Cells were lysed and sonicated in HKMG buffer (10 mM HEPES [pH 7.9], 100 mM KCl, 5 mM MgCl2, 10% glycerol, 1 mM DTT, 0.5% NP-40) containing protease and phosphatase inhibitors. After centrifugation at 12,000*g* for 10 min, protein extracts were precleared with streptavidin–agarose beads. The *ITGA1* promoter was used as a template to amplify the p53RE. PCR amplification was performed using a biotinylated forward primer and a non-biotinylated reverse primer, listed in S2 Table. Amplicons were extracted from agarose gel by using the MinElute Gel Extraction Kit (Qiagen) and quantified. Then, 2 mg of cellular protein extracts were incubated with 1 μg of biotin-ITGA1 promoter probes and 10 μg of poly(dI-dC)· poly(dI-dC) for 16 h at 4˚C. DNA-bound proteins were collected with streptavidin–agarose beads for 1 h and washed 5 times with HKMG buffer. DNA-bound proteins were then analyzed by IB.

## Electromobility shift assay

Nuclear extracts from cells were prepared as previously described [50]. Briefly, $3 \times 10^6$ cells were collected, washed in PBS 1×, and resuspended in hypotonic buffer A (10 mM HEPES [pH 7.9], 1.5 mM MgCl$_2$, 10 mM KCl, 0.5 mM DTT, 0.2 mM PMSF). The cell suspensions were then incubated on ice and homogenized by 15 passages through a 25-gauge needle. Cytoplasm fractions were collected by centrifugation at 12,000 rpm for 1 min at 4˚C. Nuclei were washed in buffer A, centrifuged, and dissolved in hypertonic buffer B (20 mM HEPES [pH 7.9], 25% glycerol, 0.42 M NaCl, 1.5 mM MgCl$_2$, 0.2 mM EDTA, 0.5 mM DTT). The nuclear extracts were collected by centrifugation at 12,000 rpm for 2 min at 4˚C. Protein concentration was estimated using an assay kit (Bio-Rad). Then, 5 μg of the extracts were incubated with 0.5 pmol of biotin-labeled DNA probe (listed in S2 Table) and poly (dI-dC) in binding buffer (10 mM Tris, 100 mM NaCl, 1 mM EDTA, 1 mM DTT, 5% glycerol [pH 7.5]) in a final volume of 15 μL. Binding reactions were incubated for 20 min at room temperature. The dye solution was then added and samples were loaded into a 5% polyacrylamide gel in 0.5× Tris-borate-EDTA buffer for running. The gels were then transferred to BM-Nylon (+) blotting membrane (Roche) and developed by using the Chemiluminescent Nucleic Acid Detection Module provided in the non-radioactive LightShift Chemiluminescent EMSA Kit (Thermo Scientific). Specificity of the protein–DNA complex was verified by a competition experiment where the nuclear extracts were incubated with an excess of unlabeled DNA.

## Sucrose gradient protein complex isolation

Sucrose density gradients were prepared from 10% to 50% sucrose (10 mM NaCl, 2 mM Tris HCl, 0.5 mM MgCl$_2$). Cells treated with 2AP or PBS:glacial acetic acid (200:1) 10 mM for 4 h were processed for nuclear extraction as previously described, and a total of 1 mg of protein was added to the sucrose gradient; 20 μg of nuclear extract was kept as input control. After 16 h of centrifugation at 35300 rpm and 4˚C, 21 fractions with a total volume of 500 μL were collected.

## Protein immunoprecipitation

IPs were performed as previously described [51]. Briefly, 5 μL of the indicated antibodies (p53DO-1, PKR, or IgG) were pre-adsorbed on 50 μL of protein A/G plus agarose beads (SC-2003) and suspended in PBS-1% NP-40 for 2 h at 4˚C. Suspended beads were also incubated with total protein lysate, nuclear protein lysate or sucrose protein fractions 5 to 21 for pre-clearing under rotation at 4˚C for 2 h. After overnight incubation with the extracts, beads were

resolved by SDS–PAGE and transferred to polyvinylidene difluoride membranes (Perkin Elmer). IB was performed with DNMT1, PKR, and p53 antibodies, and immunoreactivity was revealed by means of secondary antibodies specific for IP (Abcam). Immunoreactive proteins were visualized by means of the ECL method (Millipore).

### Exome analysis

The exome analysis was performed as described previously [32]. The SIFT missense predictions for genomes annotator was used to predict whether the amino acid substitution affects protein function [52].

### Statistical analysis

Statistical significance was determined using the Student $t$ test with Prism7 (GraphPad). The levels of statistical significance for each experiment (*, $p < 0.05$; **, $p < 0.01$; ***, $p < 0.001$; ****, $p < 0.0001$; ns, not significant) are indicated in the corresponding figures. The error bars in the graphs represent the standard deviation.

### Supporting information

**S1 Fig.** (A) Schematic representation of the *ITGA1* promoter. p53 REs were predicted using TFBind and JASPAR software. (B) 38HK cells were transfected with sense (control) and antisense oligonucleotides against ΔNp73α. After 24 h, cells were collected and processed for RT-qPCR (left) or IB (right) ($n = 3$). ns, not significant. (C) Chromatin from 38HK was processed for ChIP experiments using p73 antibody. Results were obtained by qPCR with primers spanning p53RE2 or a negative control region (nc) (S2 Table). Error bars indicate standard deviations from 2 independent experiments performed in duplicate. ns, not significant. (D) 38HK cells were treated with cyclic pifithrin-α hydrobromide or DMSO as a control for 24 h, stained with trypan blue, and counted. The histograms represent the means of 4 independent experiments. *, $p < 0.05$; **, $p < 0.01$. (E) Cells expressing N-HA-p53 or p53-C-HA were seeded into 96-well plates. After 24, 48 and 72 h, cells were incubated with 20 μL of MTS solution for 2 h. Absorbance was obtained at 490 nm. Data shown are the means of 2 independent experiments performed in duplicate. ***, $p < 0.001$; ****, $p < 0.0001$. DOI: 10.6084/m9.figshare.12732737
(TIF)

**S2 Fig. Raw data of all IB experiments were provided to the editor for review purposes and uploaded on Figshare (DOI: 10.6084/m9.figshare.12733427).**
(PDF)

**S1 Table. Sequences of siRNA and CRISPR/Cas9 vectors used for gene silencing.** DOI: 10.6084/m9.figshare.12733442.
(DOCX)

**S2 Table. Primers for RT-qPCR, ChIP and EMSA experiments.** DOI: 10.6084/m9.figshare.12733445.
(DOCX)

### Acknowledgments

We are grateful to all members of our laboratories for their cooperation. We are also grateful to Nicole Suty for her help with preparation, and Dr Karen Müller for editing this manuscript.

The authors alone are responsible for the views expressed in this article, and they do not necessarily represent the views, decisions, or policies of the institutions with which they are affiliated.

## Author Contributions

**Conceptualization:** Maria Carmen Romero-Medina, Assunta Venuti, Rosita Accardi, Massimo Tommasino.

**Formal analysis:** Maria Carmen Romero-Medina, Assunta Venuti, Alexis Robitaille.

**Funding acquisition:** Massimo Tommasino.

**Investigation:** Maria Carmen Romero-Medina, Assunta Venuti, Giusi Melita, Alexis Robitaille, Maria Grazia Ceraolo, Laura Pacini, Cecilia Sirand, Daniele Viarisio, Valerio Taverniti, Purnima Gupta, Mariafrancesca Scalise, Cesare Indiveri, Rosita Accardi.

**Methodology:** Maria Carmen Romero-Medina, Assunta Venuti, Daniele Viarisio, Rosita Accardi, Massimo Tommasino.

**Project administration:** Massimo Tommasino.

**Resources:** Daniele Viarisio, Cesare Indiveri, Massimo Tommasino.

**Supervision:** Massimo Tommasino.

**Validation:** Maria Carmen Romero-Medina, Assunta Venuti, Rosita Accardi, Massimo Tommasino.

**Visualization:** Maria Carmen Romero-Medina, Assunta Venuti, Massimo Tommasino.

**Writing – original draft:** Massimo Tommasino.

**Writing – review & editing:** Maria Carmen Romero-Medina, Assunta Venuti, Massimo Tommasino.

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
