## [Decision Letter · Decision Letter 0]

15 Apr 2020

Dear Dr Tommasino,

Thank you very much for submitting your manuscript "Human papillomavirus type 38 alters wild-type p53 activity to promote cell proliferation via the downregulation of integrin alpha 1 expression" for consideration at PLOS Pathogens. As with all papers reviewed by the journal, your manuscript was reviewed by members of the editorial board and by several independent reviewers. In light of the reviews (below this email), we would like to invite the resubmission of a significantly-revised version that takes into account the reviewers' comments.

We cannot make any decision about publication until we have seen the revised manuscript and your response to the reviewers' comments. Your revised manuscript is also likely to be sent to reviewers for further evaluation.

Sincerely,

Paul Francis Lambert

Associate Editor

PLOS Pathogens

Alison McBride

Section Editor

PLOS Pathogens

Kasturi Haldar

Editor-in-Chief

PLOS Pathogens

orcid.org/0000-0001-5065-158X

Michael Malim

Editor-in-Chief

PLOS Pathogens

orcid.org/0000-0002-7699-2064

Reviewer's Responses to Questions

**Part I - Summary**

Reviewer #1: Romero-Medina et al. describe the ability of HPV38 E6 proteins to alter p53 phosphorylation through effects on DNMT1 and PKR leading to suppression of integrin A1 (ITGA1) expression which in turn is critical for cell growth. Previous studies from this group showed that HPV38 E6 and E7 can immortalize human keratinocytes through alteration in p53 and Rb. The present study show this results in suppression of expression of integrin alpha 1 and is mediated by E6. The IGTA1 promoter has two p53 reactive sites that are bound by p53 together with DNMT1 , a gene silencing factor. Additional work shows the S392 phosphorylated isoform of p53 is the primary form bound and that its phosphorylation is mediated by PKR which is also regulated by E6. Furthermore PKR, DNMT1 and p53 form trimeric complexes that can be isolated on sucrose gradients. Overexpression of IGTA1 in HPV 38 positive cells resulted in decreased colony formation and cell death which correlated with changes in EGFR activity. Overall this is an interesting study examining the role of integrin alpha 1 in regulating proliferation of HPV 38 positive cells.

Reviewer #2: Romero-Medina and colleagues present an interesting study on a potential role of wt p53 in promoting cell proliferation in the context of HPV38 E6/E7 containing keratinocytes. This appears to be linked to downregulation of integrin alpha expression and increased levels of EGFR signalling. Mechanistically this is linked to E6/E7 stimulation of PKR activity which in turn phosphorylates p53 and confers interaction with DNMT1 resulting in suppression of the integrin alpha promoter. This is a very interesting study and reveals a novel role for wt p53 in promoting cell proliferation. There are however several experiments still required to fully support the conclusions.

Reviewer #3: The study by Romero et al builds on previous work from the Tommasino group demonstrating the manipulation of p53 function by beta-HPV E6/E7 oncoproteins to promote carcinogenesis. The study is mechanistic, well written and generally a pleasure to read. I have a few relatively minor comments that should be corrected but it is my opinion that this study is of the quality required for publication in PLoS Pathogens.

Sentence in lines 65-67 does not make sense and needs editing.

Line 101 - sentence needs editing.

The downregulation of ITGA1 protein by E6/E7 expression is demonstrated by immunofluorescence. This is acceptable although I wonder why they did not show this by Western blotting. Nonetheless, these IF data would be strengthened by quantification of multiple images which can be easily achieved using ImageJ.

The binding of DNMT1 to the p53-RE2 region of the ITGA1 promoter shown by oligo pull-down in figure 2C is not very convincing. The conclusion would be strengthened by quantification of multiple pull down experiments to show consistency.

Why does CRISPR/Cas9 knockout of p53 only result in a small decrease in p53 mRNA abundance (Fig 3B). This reduction does reach significance but I would have expected a more dramatic decrease.

How specific is the T446-PKR antibody used in the ChIP-reChIP assay shown in figure 5C and WB shown in figure 5D? Please provide a reference for a study demonstrating specificity.

The authors state that the sucrose fractionation shown in Figure 5D demonstrates a trimeric complex containing p53/PKR/DNMT1. While the association between PKR and p53 is suggested by this experiment, the association of DNMT1 is less clear. Indeed, the treatment of cells with PKR inhibitor 2AP does not obviously alter DNMT1 fractionation. The interaction between p53 and DNMT1 is not clearly shown in the present study and I do not see sufficient evidence to conclude that this interaction is regulated by PKR phosphorylation of p53 on S392 as stated in lines 211-212.

The flow cytometry data shown in figure 6B are not clear – the histograms are too pale and hard to see.

The Western blot shown in Figure 7E should be improved – at present there is no clear separation between the bands and it is therefore difficult to see the sequential reduction in HA-p53 expression following TMX treatment.

**Part II – Major Issues: Key Experiments Required for Acceptance**

Reviewer #1: no major issues

Reviewer #2: Specific Points.

1. Figure 2 and onwards. All assays throughout are performed in HPV 38 positive cells. Interaction assays and CHIP assays should also be done for comparison in HPV negative cells to show clear stimulation by HPV38 E6/E7.

2. Figure 2C. The DNMT signal is marginal. What happens if cells are transfected with siRNA p53 or siRNA DNMT. This is alluded to later but it would be good to know which protein is recruiting which?

3. What happens with the band shift assays if phospho-mimic S392 p53 is included?

4. Fig.S1B. This is critical data as all previous work has focussed on the role of DeltaNp73 – relevant siRNA approaches should be shown.

5. Figure 3. This should have HPV negative cells for comparison.

6. Figure 4. What happens if mutant S392A p53 is included in these assays – and/or the phospho mimic. As the data stands it shows that phosphorylated S392 is bound but it does not show that it is phosphorylation at that specific site which is responsible. Is activation of PKR due to E6 or E7 or both? Assays using cells expressing them singly would be useful.

7. Figure 5A. The co.ip data is weak. Again mutants of p53 at S392 would help clarify the molecular details.

8. Throughout there appears to be two PKR bands on the blots – which is the bona fide protein. A simple siRNA should be performed to clarify which band is PKR.

9. Figure 5B. I don’t understand why the pS392 form of p53 co.ips better than total protein – I would have thought the phospho form would no longer be bound. Again mutants of p53 would help clarify what is occurring here.

10. Figure 5D. This is critical to the whole story – a phosphomimic/phospho destroyed is really essential to showing that it is phosphorylation at that site which is critical.

Reviewer #3: See above

**Part III – Minor Issues: Editorial and Data Presentation Modifications**

Reviewer #1: 1). What is known about the effects of IGTA1 in high-risk HPV 16 or 18 positive cells? E6 reduces p53 levels in these cells so IGTA1 should go up yet cells do not exhibit reduced colony formation. This is puzzling and needs to be discussed.

2). What role could IGTB1 which is also increased by HPV38 proteins?

3). How does overexpression of IGTA1 compare to the levels seen in normal keratinocytes? What are the increases in p53 in HPV 38 cells compared to normal keratinocytes?

4). How does the observation with the transgenic mice with the third cSCCC lesion that had increased ICTA1 and non-deleterious EGFR mutations fit in with their model for effects of IGTA1? Is p53 mutated in these cells?

5). Section around line 236: What is the nature of the mutation that is introduced into the transduced p53 and how does this address mechanism? Is this mutant still able to bind DNA and transactivate? This section is confusing and needs to be clarified

6). Figure 3B: CRISPR-p53 only reduces p53 levels by 50% which is not very efficient. Why is it so inefficient?

Reviewer #2: (No Response)

Reviewer #3: See above

PLOS authors have the option to publish the peer review history of their article (what does this mean?). If published, this will include your full peer review and any attached files.

Reviewer #1: No

Reviewer #2: No

Reviewer #3: No
---

## [Decision Letter · Decision Letter 1]

8 Jul 2020

Dear Dr Tommasino,

We are pleased to inform you that your manuscript 'Human papillomavirus type 38 alters wild-type p53 activity to promote cell proliferation via the downregulation of integrin alpha 1 expression' has been provisionally accepted for publication in PLOS Pathogens.

Best regards,

Paul Francis Lambert

Associate Editor

PLOS Pathogens

Alison McBride

Section Editor

PLOS Pathogens

Kasturi Haldar

Editor-in-Chief

PLOS Pathogens

orcid.org/0000-0001-5065-158X

Michael Malim

Editor-in-Chief

PLOS Pathogens

orcid.org/0000-0002-7699-2064

Reviewer Comments (if any, and for reference):

Reviewer's Responses to Questions

**Part I - Summary**

Reviewer #1: THe authors have addressed most issues raised in the initial review and is suitable for publication

Reviewer #2: (No Response)

Reviewer #3: The authors have addressed all of my initial concerns with additional data included or editing of the text.

**Part II – Major Issues: Key Experiments Required for Acceptance**

Reviewer #1: none

Reviewer #2: The authors have done a good job in answering most of the comments. However the key issue of whether the effects are due to phosphorylation at the specific S392 residue remain. As it stands all the authors can state is that phosphorylation of p53 is important - but without the appropriate mutants the molecular basis for these observations remains obscure.

I realise there are problems in accessing labs at the moment but the generation of the appropriate S392A and S392E should have been done much earlier in the study to provide this key mechanistic information.

Reviewer #3: (No Response)

**Part III – Minor Issues: Editorial and Data Presentation Modifications**

Reviewer #1: none

Reviewer #2: (No Response)

Reviewer #3: (No Response)

PLOS authors have the option to publish the peer review history of their article (what does this mean?). If published, this will include your full peer review and any attached files.

Reviewer #1: No

Reviewer #2: No

Reviewer #3: No

---

## [Editor Report · Acceptance letter]

12 Aug 2020

Dear Dr Tommasino,

We are delighted to inform you that your manuscript, "Human papillomavirus type 38 alters wild-type p53 activity to promote cell proliferation via the downregulation of integrin alpha 1 expression," has been formally accepted for publication in PLOS Pathogens.

Best regards,

Kasturi Haldar

Editor-in-Chief

PLOS Pathogens

orcid.org/0000-0001-5065-158X

Michael Malim

Editor-in-Chief

PLOS Pathogens

orcid.org/0000-0002-7699-2064